# Exploring Brazilian Green Propolis Phytochemicals in the Search for Potential Inhibitors of B-Raf^600^E Enzyme: A Theoretical Approach

**DOI:** 10.3390/ph18060902

**Published:** 2025-06-16

**Authors:** Garcia Ferreira de Souza, Airis Farias Santana, Fernanda Sanches Kuhl Antunes, Ramon Martins Cogo, Matheus Dornellas Pereira, Daniela Gonçales Galasse Rando, Carolina Passarelli Gonçalves

**Affiliations:** 1Biological Science Department, Instituto Universitario Italiano de Rosario, Corrientes 720, PB y 1er Piso Rosario S2000CTT, Santa Fe, Argentina; garciafdesouza@gmail.com (G.F.d.S.); iris.farias@icloud.com (A.F.S.); 2Cosmobeauty, Avenida Aruanã 1160, Barueri 06460-010, SP, Brazil; fernanda.sanches.dra@gmail.com; 3Postgraduate Program in Chemical Biology, Universidade Federal de São Paulo, Diadema 09913-030, SP, Brazil; ramon.cogo@unifesp.br (R.M.C.); matheus.dornellas16@unifesp.br (M.D.P.); dgrando@unifesp.br (D.G.G.R.)

**Keywords:** Brazilian green propolis, Artepillin C, B-Raf^600^E, melanoma, molecular docking, molecular dynamics, natural products, MAPK signaling pathway, selective inhibition, in silico drug discovery

## Abstract

**Background/Objectives:** Melanoma is one of the most aggressive forms of skin cancer and is frequently associated with the B-Raf^600^E mutation, which constitutively activates the MAPK signaling pathway. Although selective inhibitors such as Vemurafenib offer clinical benefits, their long-term efficacy is often hindered by resistance mechanisms and adverse effects. In this study, twelve phytochemicals from Brazilian green propolis were evaluated for their potential as selective B-Raf^600^E inhibitors using a computational approach. **Methods:** Physicochemical, ADME, and electronic properties were assessed, followed by molecular docking using the B-Raf^600^E crystal structure (PDB ID: 3OG7). Redocking validation and 500 ns molecular dynamics simulations were performed to investigate the stability of the ligand-protein complexes, and free energy calculations were then computed. Results: Among the tested compounds, Artepillin C exhibited the strongest binding affinity (−8.17 kcal/mol) in docking and maintained stable interactions with key catalytic residues throughout the simulation, also presenting free energy of binding ΔG of −20.77 kcal/mol. HOMO-LUMO and electrostatic potential analyses further supported its reactivity and selectivity. Notably, Artepillin C remained bound within the ATP-binding site, mimicking several critical interactions observed with Vemurafenib. **Results:** Among the tested compounds, Artepillin C exhibited the strongest binding affinity (−8.17 kcal/mol) and maintained stable interactions with key catalytic residues throughout the simulation. HOMO-LUMO and electrostatic potential analyses further supported its reactivity and selectivity. Notably, Artepillin C remained bound within the ATP-binding site, mimicking several critical interactions observed with Vemurafenib. **Conclusions:** These findings indicate that Artepillin C is a promising natural compound for further development as a selective B-Raf^600^E inhibitor and suggest its potential utility in melanoma treatment strategies. This study reinforces the value of natural products as scaffolds for targeted drug design and supports continued experimental validation.

## 1. Introduction

Melanoma is one of the most aggressive and deadly forms of skin cancer, notorious for its rapid progression and resistance to traditional treatments [1]. Understanding the molecular drivers behind its growth has led to significant breakthroughs in targeted therapies. Among the key players in melanoma development is the enzyme B-Raf, particularly its mutated form known as B-Raf^600^E [1,2]. This mutation occurs when valine at position 600 is replaced by glutamic acid, causing the enzyme to become permanently active. As a result, the MAPK signaling pathway, which controls cell growth and survival, is continuously turned on, leading to uncontrolled cell division and tumor growth. The discovery of B-Raf^600^E’s role in melanoma was a turning point, highlighting it as a crucial target for drug development [3]. Researchers have been focusing on designing inhibitors that could specifically bind to this mutant form without affecting the non-mutated form of the protein, thus minimizing side effects. Vemurafenib was one of the first successful drugs to achieve this [2]. Although Vemurafenib represented a significant advance in the treatment of melanoma with the B-Raf^600^E mutation, its long-term efficacy is often hindered by important limitations.

One of the most pressing challenges is the rapid development of resistance. While many patients initially respond favorably, tumors frequently acquire adaptive mechanisms that reactivate the MAPK signaling pathway, the very cascade that Vemurafenib aims to inhibit [4]. Beyond resistance, Vemurafenib is also associated with several adverse effects, particularly dermatological toxicities. Commonly reported reactions include photosensitivity and, in some cases, the development of cutaneous squamous cell carcinomas. Arthralgia is another frequent side effect, which can significantly impact the patient’s quality of life and may necessitate dose adjustments or even treatment discontinuation [5]. Given these limitations, ongoing research has increasingly focused on developing next-generation therapies, including more selective BRAF inhibitors and combination regimens, with the goal of enhancing efficacy while minimizing toxicity and delaying the onset of resistance [6]. In this context, natural extracts have also been investigated as complementary or alternative therapeutic strategies for melanoma.

Among those showing promising results is Brazilian green propolis extract, which has demonstrated antitumor, pro-apoptotic, and anti-angiogenic effects in preclinical studies, suggesting potential as an adjunct in melanoma treatment [7,8,9,10,11,12]. Due to the encouraging outcomes observed with Brazilian green propolis, further studies are warranted to isolate and identify the specific bioactive compounds responsible for these effects. Such investigations could help determine which substances have the greatest therapeutic potential and, importantly, whether they are capable of selectively inhibiting the mutant B-Raf^600^E, offering a novel and potentially less toxic approach to melanoma treatment. Brazilian green propolis is a resinous substance collected by *Apis mellifera* bees from the buds and exudates of plants, especially *Baccharis dracunculifolia*. This propolis is widely recognized for its biological properties, including antioxidant, antibacterial and antiproliferative activities [13]. Studies have shown that Brazilian green propolis has a distinct chemical composition, rich in prenylated derivatives of *p*-coumaric acid and flavonoids, especially Artepillin C, which is found in high concentration in this variety of propolis [13,14]. In addition to *p*-coumaric acid (**1**) and Artepillin C (**3**), its main chemical marker, several other compounds have been identified in Brazilian green propolis, including drupanin (**2**) [15], culifolin (**4**) [16], ferulic acid (**5**) and caffeic acid (**6**) [17], kaempferol (**7**) and kaempferide (**8**) [18], ermanin (**9**) [19], (*E*)-2-isopropenyl-7-isopentenyl-2,3-dihydrobenzofuran-5-acrylic acid (**10**), 2,2-dimethyl-2*H*-1-benzopyran-6-carboxylic acid (**11**) and (*E*)-3-(2,2-dimethyl-2*H*-1-benzopyran-6-yl) propenoic acid (**12**) [20]. The compounds selected for this study represent the main chemical constituents of Brazilian green propolis, including both major markers, such as *p*-coumaric acid and Artepillin C, and secondary bioactive compounds. The selection prioritized compounds are representative of the chemical diversity of green propolis, which has a history of description in the literature and potential biological activity, allowing a more comprehensive analysis of their structural correlations and relevant properties for the development of medicinal formulations. In this preliminary study, these compounds were evaluated for their potential as selective inhibitors of the B Raf^600^E mutant protein using bioinformatics and cheminformatics approaches. The investigation focused on structural and conformational analyses, employing molecular modeling techniques such as molecular docking to predict binding affinities and interaction modes, as well as molecular dynamics simulations to assess the stability and behavior of the ligand-protein complexes over time.

## 2. Results

### 2.1. Physicochemical Properties and ADME Studies

The physicochemical properties of twelve phytochemicals present in Brazilian green propolis were calculated theoretically using the SWISSADME platform, with a focus on their possible drug-like characteristics. The molecular properties analyzed included molecular weight (MW), number of heavy atoms, number of aromatic heavy atoms, fraction of sp^3^ hybridized carbons (Csp^3^ fraction), number of rotational bonds, number of hydrogen bond acceptors (HBAs), number of hydrogen bond donors (HBDs), molar refractivity (MR) and topological polar surface area (TPSA). These parameters are essential for predicting the pharmacokinetic behavior of the compounds, such as solubility, permeability and bioavailability. The results are shown in Table 1.

The pharmacokinetic parameters of the evaluated phytochemicals were assessed to predict their ADME behavior. All ligands demonstrated high gastrointestinal (GI) absorption, indicating favorable oral bioavailability. The blood–brain barrier (BBB) permeability was observed for most compounds, except for ligands **6**, **7**, **8**, and **9**. None of the ligands were identified as substrates for *p*-glycoprotein (P-gp), suggesting a low potential for efflux-mediated drug resistance. The LogP values ranged from 0.93 to 4.27, with the majority falling between **1** and **3**. Notably, ligands **3**, **4**, and **10** had values greater than **4**. Regarding cytochrome P450 enzyme inhibition, ligands **1**, **2**, and **5** showed no inhibitory activity against any of the tested isoenzymes. Ligand **3** inhibited CYP3A4, while ligand **4** inhibited both CYP3A4 and CYP2D6. Ligand **10** inhibited CYP1A2, CYP2D6, and CYP3A4. Ligands **7**, **8**, and **9** inhibited CYP2C19, CYP2C9, CYP2D6, and CYP3A4. These results are summarized in Table 2.

### 2.2. Molecular Modeling

#### 2.2.1. Geometry Optimization

To obtain the most stable conformations of the major phytochemicals identified in Brazilian green propolis, a hybrid conformational optimization protocol was applied. This approach combined three levels of molecular modeling: initial geometry optimizations were performed using classical molecular mechanics with the Universal Force Field (UFF), followed by semi-empirical calculations using the AM1 method, and finally refined using an ab initio Hartree–Fock (HF/6-31G*) level of theory. This stepwise strategy aimed to maximize accuracy by combining the efficiency of classical methods with the precision of quantum chemical approaches. The optimized 3D structures are presented in Figure 1. Among the selected phytochemicals, compounds **3** and **10** exhibited the highest degree of conformational flexibility, each containing six rotatable bonds. These structural features indicate a greater potential for dynamic adaptation during molecular recognition. The conformational planarity of compound **7** (a flavonol with hydroxyl groups at positions 3, 5, 7, and 4′) was also notable, suggesting enhanced intramolecular conjugation. In contrast, methoxylated derivatives, such as compounds **8** and **9**, showed a decreased planarity and increased hydrophobicity due to substitutions on the B and C rings.

#### 2.2.2. HOMO and LUMO Orbitals

The highest occupied (HOMO) and lowest unoccupied (LUMO) molecular orbitals, known as frontier orbitals, are important for understanding the electrical and optical properties and reactivity of molecules. The energy difference between HOMO and LUMO, known as the energy gap, is particularly important as it directly influences the chemical reactivity of the molecule. Compounds with smaller energy gaps tend to be more reactive, which can facilitate interaction with the biological target, while those with larger gaps have greater chemical stability, allowing for more controlled interactions. In addition, frontier orbitals are involved in electron transfer processes, such as oxidation and reduction reactions, which are often essential for the biological activity of active molecules [21]. The graphical depiction of the HOMO and LUMO orbitals for the compounds studied is presented in Figure 2.

#### 2.2.3. Electrostatic Potential Map (EPM)

The electrostatic potential map (EPM) (Figure 3) is a valuable tool for understanding how molecules interact with biological targets. It visualizes the distribution of electrostatic charges across the molecular surface, allowing for the identification of regions most likely to engage in interactions that influence binding affinity and selectivity. This information is crucial for optimizing the therapeutic potential of bioactive compounds. Electron-rich (highly electronegative) regions often act as hydrogen bond acceptors, while electron-deficient (electropositive) regions typically function as hydrogen bond donors [22]. Beyond predicting molecular interactions, EPMs also contribute to evaluating pharmacokinetic properties such as solubility and membrane permeability. While highly polar regions can enhance aqueous solubility, they may simultaneously hinder passive diffusion through biological membranes, ultimately affecting bioavailability. Furthermore, mapping the electrostatic profile of a molecule provides insights into its metabolic stability by highlighting areas susceptible to enzymatic transformations, factors that can impact efficacy or lead to the formation of toxic metabolites [23].

### 2.3. Molecular Docking

The B-Raf^600^E enzyme is a key regulator of the MAPK signaling pathway, playing a central role in controlling cell growth and survival. Its structure is finely tuned to support its function as a serine/threonine kinase. The enzyme comprises two principal lobes: a smaller N-terminal lobe, composed mainly of β-sheets, and a larger C-terminal lobe, rich in α-helices. These lobes are connected by an ATP-binding cleft—the catalytic core where phosphorylation of substrate proteins occurs. This region is especially critical, as it positions ATP for phosphate transfer during kinase activity. In the B-Raf^600^E mutant, a single amino acid substitution—valine to glutamic acid at position 600—has significant structural and functional consequences. Valine is a small, nonpolar residue, while glutamic acid is bulkier and carries a negative charge. This alteration destabilizes the inactive conformation of the kinase, favoring a constitutively active state even in the absence of upstream signals. The result is continuous MAPK pathway activation, driving uncontrolled cell proliferation and contributing to melanoma development. Structural analysis, particularly based on the 3OG7 crystal structure, reveals how this mutation reconfigures the ATP-binding pocket, creating new opportunities for selective drug binding. Vemurafenib, a potent B-Raf^600^E inhibitor, binds precisely within this pocket, forming a network of hydrogen bonds and hydrophobic interactions. It locks the enzyme in an inactive conformation, effectively preventing ATP binding and blocking downstream signaling.

The Glu600 residue, in combination with adjacent positively charged residues, helps shape a unique hydrophobic pocket that Vemurafenib exploits, explaining its high selectivity for the mutant over the wild-type enzyme—thus minimizing off-target effects. These structural insights have been instrumental in the development of targeted melanoma therapies. However, despite the initial success of B-Raf inhibitors, resistance often emerges through secondary mutations or alternative activation of the MAPK pathway. To address this challenge, combination therapies—such as co-administering B-Raf and MEK inhibitors—have been investigated to enhance efficacy and delay resistance. In this context, a molecular docking approach was employed to investigate the potential interactions between phytochemical ligands from Brazilian green propolis and the active site of the B-Raf^600^E mutant. The docking results, which provide insights into binding affinities and interaction profiles, are summarized in Table 3.

### 2.4. Molecular Dynamics and Free Energy of Binding

Molecular dynamics (MD) simulations were performed for 500 nanoseconds (ns), starting from the most favorable ligand-enzyme complexes identified through molecular docking. To enable comparative analysis, simulations of the free enzyme and Vemurafenib were also conducted under the same theoretical conditions described in the “Molecular Dynamics” section compound **3**. Artepillin C remained stably bound in the active site throughout the entire 500 ns simulation. Moreover, the protein exhibited enhanced structural stability from 200 ns onward, with the Root Mean Square Deviation (RMSD) stabilizing around 0.05 nm. The radius of the gyration profile also revealed a consistent compaction pattern, which means that the structural integrity of the protein remained preserved throughout the whole simulation, as shown in Figure 4. In the absence of the ligand (Figure 4, red lines), the enzyme appeared more compact until 300 ns but less stable, also expanding from 300 ns onwards. This suggests that Artepillin C contributes to the structural stabilization of the enzyme. The same patterns seen for Artepillin C are observed on Vemurafenib MD. The radius of gyration starts with contraction followed by relaxation from 300 ns onwards. On the other hand, for RMSD, Vemurafenib presents a less stabilizing capability than Artepillin C, with a delta of 1 nm from 200 ns to the end of the simulation. This variability is closer to the free enzyme RMSD variation, meaning Artepillin C is a better stabilizer of the protein.

The Artepillin C hydrogen bond (hbond) analysis revealed that the complex was maintained by two persistent hbonds throughout the simulation, occasionally reaching 3 hbonds. Vemurafenib yet presents a crescent number of hbond interactions along the simulation, preserving seven hbonds from 200 ns onward. Additionally, analysis of the most representative pose from the simulation confirmed that Artepillin C remained bound to the ATP-binding site even after 300 ns (Figure 5).

Compared to the crystal structure, the cavity appeared more open due to system relaxation, yet the compound occupied regions associated with Vemurafenib’s binding mode, especially near PHE583 (Figure 6). Interaction analysis from the most representative pose in the clusters revealed two key hydrogen bonds with ASP594 and a critical ion–ion interaction between the carboxylate group of Artepillin C and LYS473 of B-Raf (Figure 7A,C). Hydrophobic interactions were also observed, involving Artepillin’s prenyl branches and apolar residues such as LEU514, VAL471, ALA481, LYS483 and ILE463. Other surrounding residues also contributed via potential van der Waals interactions, as represented in Figure 7C. The hydrophobic surface map confirmed that the prenyl branches oriented toward hydrophobic regions of the pocket (Figure 7B), while the carboxylate group projected outward to interact with polar regions and solvent-accessible water molecules. The hydroxyl group was located deeper within the binding site, interacting with ASP594.

As shown in Table 4, the major energy contributions for both Artepillin C and Vemurafenib come from electrostatic (EEL) and Van der Waals (VDW) components. However, ENPOLAR, which is also associated with hydrophobic interaction along with VDW, provides a minor contribution to enthalpic energy in both complexes. These data reinforce what was already observed during interaction analysis, i.e., for Artepillin C, the hydrophobic and VDW interactions observed along MD for ILE463, VAL471 and ALA481 are confirmed. Additionally, ASP594 is identified as an unfavorable interaction due to a desolvation penalty. For Vemurafenib, CYS532, GLN530, VAL471, ALA481, PHE583, PHE595 present modest contributions.

Furthermore, PHE595 also contributes unfavorable interaction due to desolvation penalty (Figure 8). Regarding affinities, Vemurafenib presents a lower ΔH of −60.25 kcal/mol than Artepillin C, which has a ΔH of −22.05 kcal/mol. This difference can be associated with the quantity of interactions observed for Vemurafenib, as shown in Figure 5 and Figure 8. With respect to the entropy contribution, Vemurafenib displays a higher value. Although this entropic loss can be interpreted as thermodynamically favorable due to the associated gain in disorder, its resulting free energy is higher than that of Artepillin C. This difference suggests that the overall thermodynamic stability of Vemurafenib is lower, making Artepillin C more spontaneously favorable.

## 3. Discussion

### 3.1. Physicochemical Properties and ADME Studies

The molecular weights (MW) of the studied compounds ranged from 164.16 to 314.29 g/mol, with ligand **1** having the lowest MW and ligand **9** the highest. All compounds fall within the acceptable range for drug-like molecules (<500 g/mol), suggesting favorable pharmacokinetic properties. Aromatic heavy atoms were prominent across the series, varying from 6 to 16, contributing to planar and rigid structures. The fraction of sp^3^ hybridized carbons (Fraction Csp^3^), which indicates molecular complexity and three-dimensionality, varied among the compounds. Ligands **1** and **6** exhibited a fraction Csp^3^ of 0, indicating completely planar structures, while compound **3** showed the highest value (0.32), suggesting increased structural diversity. Hydrogen bonding capacity, which affects solubility and permeability, was evaluated through hydrogen bond acceptors (HBAs) and donors (HBDs). The HBA values ranged from 3 to 6, while HBDs varied from **1** to **4.** Molecules **7**, **8**, and **9** showed the highest numbers of both HBAs and HBDs, consistent with their polyphenolic structures. These molecules also exhibited the highest topological polar surface area (TPSA), with ligand **7** having the maximum value of 111.13 Å^2^, increasing solubility but may limit passive membrane permeability.

Molar refractivity (MR) ranged from 45.13 in ligand **1** to 92.57 in ligand **3**, correlating with molecular size and complexity. The extensive conjugation present in these molecules suggests a strong potential for interactions with aromatic residues in protein targets, potentially enhancing binding affinity. Overall, these findings provide a detailed overview of the physicochemical properties influencing the drug-likeness of the studied molecules. The balance between molecular weight, aromaticity, hydrogen bonding capacity, and structural flexibility suggests promising potential for these compounds as medicinal candidates. Molecules with moderate MW, balanced TPSA, and optimal Fraction Csp^3^, like ligand **3**, exhibit the most favorable profile for further development.

The high GI absorption values observed across all compounds support their potential as orally active agents. Most ligands’ ability to cross the BBB suggests possible central nervous system activity. The lack of P-gp substrate characteristics indicates a lower risk of drug resistance via efflux mechanisms. The range of LogP values suggests generally favorable mem-brane permeability, while compounds with higher values may benefit from increased BBB penetration. The distinct CYP inhibition profiles highlight varying degrees of potential metabolic interaction. Compounds with minimal or no CYP inhibition, such as Ligands **1**, **2**, and **5**, are less likely to participate in drug-drug interactions. In contrast, Ligands **7**, **8**, and **9** exhibited broad-spectrum CYP inhibition, which may influence the metabolism of co-administered drugs and necessitate caution in further development. These findings underscore the importance of early pharmacokinetic screening in guiding compound selection for drug discovery.

### 3.2. Molecular Modeling

#### 3.2.1. Geometry Optimization

The main phytochemicals already identified in Brazilian green propolis are derivatives of simple or prenylated cinnamic acids and flavonoids. Cinnamic acids have a basic structure composed of an aromatic ring linked to an unsaturated aliphatic chain (CH=CH–COOH). The presence of the double bond in the side chain (CH=CH) introduces geometric isomerism (E/Z), with the E isomer generally being more stable due to the lower steric repulsion between the substituent groups. Although Z isomers are less common, they can be more reactive due to the functional proximity between the groups [24]. The single bond between the aromatic ring and the vinyl group (CH=CH–), as well as the C–C bond between the side chain and the carboxylic acid (–COOH), are rotatable, creating different conformations that influence molecular interactions, such as the formation of hydrogen bridges or π–π interactions [25]. In addition, these bonds can generate intramolecular interactions, such as hydrogen bonds, which alter the conformational stability and polarity of the compound [26]. In addition to Lipinski’s Rule of Five [27], other criteria, such as Veber’s Rule [28] and Muegge’s Rule [29] suggest that small bioactive molecules should not exceed 10 and 15 rotatable bonds, respectively, to ensure favorable pharmacokinetic properties.

It is well known that compound **3**, the majority compound present in Brazilian green propolis, is one of the most active compounds ever reported [30,31,32]. However, it’s reported in the literature as an unstable compound, and its instability is mainly attributed to the reactivity of the prenyl group and phenol with some bioactive properties, such as antioxidant activity or the ability to interact with biological targets [33,34,35]. The conformational flexibility of flavonoids is directly related to the characteristics of each ring and the interactions between them, and that determines the overall conformation of the molecule, an essential characteristic for its biological activity [36,37]. The presence of hydroxyl groups also increases the ability to form hydrogen bonds, both intra- and intermolecular, which contributes to structural stabilization [38]. In addition, methoxylation can cause a small increase in the angle between the A and B rings, slightly decreasing the planarity in relation to compound **7** [39]. Double methoxylation confers greater hydrophobicity, reduces polarity and the ability to form hydrogen bonds, as well as increases volume and conformational flexibility around the C ring [40].

#### 3.2.2. HOMO and LUMO Orbitals

The HOMO-LUMO gap values obtained for the Brazilian green propolis phytochemicals considered for this study range from 0.09142 to 0.12032 eV, reflecting the electronic diversity of these natural compounds. Their moderate ΔE values suggest an ideal balance between reactivity and stability, allowing efficient interactions with biomolecular targets such as bacterial cell walls and inflammatory mediators. The HOMO-LUMO gap values observed follow the general trends reported for bioactive polyphenols, flavonoids and prenylated phenolics, reinforcing the medicinal potential of the phytochemicals in Brazilian green propolis. Compounds such as compound **9** (0.09142 eV), compound **8** (0.09644 eV), and compound **6** (0.09649 eV) showed the lowest ΔGAP HOMO-LUMO values, suggesting greater electronic reactivity. This characteristic is often associated with high antioxidant potential, as these molecules can donate electrons to neutralize reactive oxygen species (ROS), recognized for their antioxidant and cytoprotective properties [41,42,43,44].

#### 3.2.3. Electrostatic Potential Map (EPM)

Analysis of the electrostatic potential maps revealed that all cinnamic acid derivatives exhibit a pronounced negative electronic charge on the carbonyl oxygen of the carboxylic acid group. This high electron density suggests a strong capacity for hydrogen bond acceptance, which could contribute to their known bioactivities, such as antioxidant and anti-inflammatory effects. The localization of the lowest electronic density on the phenolic group indicates potential sites for electrophilic attack, which may play a role in their radical scavenging activity. These electronic characteristics align with previous studies [45] that link the antioxidant properties of cinnamic acids to the electron-withdrawing nature of the carboxyl group, which stabilizes phenoxyl radicals formed during redox reactions. In contrast, the flavonoids **7**, **8** and **9** did not exhibit prominent regions of negative electrostatic potential. Instead, the low electronic density was localized on the phenolic group in the A ring. This suggests a distinct electronic distribution compared to the cinnamic acids, potentially influencing their interaction with biological targets. The absence of highly negative regions may reduce the strength of hydrogen-bonding interactions compared to the cinnamic acid derivatives. However, the conjugated π-electron system in the flavonoid structures could enhance π–π stacking interactions with proteins or nucleic acids, contributing to their broad spectrum of biological activities, including anti-inflammatory and anticancer properties [46]. Overall, the observed electrostatic potential patterns provide a molecular basis for understanding the diverse biological activities of these compounds. The distinct electronic characteristics of cinnamic acids and flavonoids could be correlated with their different mechanistic pathways in antioxidant, anti-inflammatory, and anticancer activities. Subsequently, our studies involving molecular docking and dynamic simulations further demonstrate the interactions between these compounds and their respective biological targets, B-RAF600, providing a more comprehensive understanding of their pharmacological potential.

### 3.3. Molecular Docking

The molecular docking simulation of twelve chemical compounds in the B-Raf^600^E enzyme revealed intriguing interactions within the active site. Critical amino acids in the active site, including GLN530, CYS532, PHE595, and GLY596, play significant roles in ligand binding and catalytic activity. The compounds exhibited varied calculated binding energies and interaction profiles, shedding light on their potential as inhibitors. Compound 3 also demonstrated a significant binding affinity of −8.17 kcal/mol, interacting through hydrogen bonds with LYS483, CYS532, and GLN530. These interactions occur near key catalytic residues, possibly hindering ATP binding. Notably, hydrophobic contacts with VAL471, LEU505, and PHE595 suggest a stable docking pose within the active site. The binding pattern reflects a strategic occupation of the hydrophobic pocket, indicative of selective inhibition potential. This compound’s interaction profile aligns with its previously reported bioactivity, supporting its continued investigation as a therapeutic agent. Other compounds, such as compound 10 and compound 4, also displayed favorable binding energies of −7.74 and −7.57 kcal/mol, respectively, but much lower than the ones observed with compound 3 and compound 12. Both compounds interacted with CYS532 and PHE595 as well as through hydrophobic interactions, particularly with residues like VAL471 and ALA481. Compound 2, compound 7, compound 8, and compound 9 presented moderate binding affinities ranging from −7.02 to −7.38 kcal/mol. Their interactions were characterized by hydrogen bonds with CYS532 and GLN530, along with hydrophobic contacts involving TRP531 and PHE583. These compounds consistently engaged with critical residues within the ATP-binding pocket; however, their binding energies, although favorable, were lower compared to compound 3 and compound 12, suggesting they could not be good ligands of the target. 

Conversely, cinnamic acid derivatives like compounds 1, 5, and 6 showed the lowest binding energies, ranging from −5.11 to −5.51 kcal/mol. Although they formed hydrogen bonds with essential residues such as CYS532 and GLN530, their hydrophobic interactions were limited. This lack of extensive hydrophobic contact might contribute to their reduced binding affinities. Their weaker interaction profiles suggest a lower potential for effective kinase inhibition. The results of the molecular docking simulation highlight several positive aspects. However, some limitations must be considered. Compounds with lower binding energies, despite forming hydrogen bonds, lacked extensive hydrophobic interactions, potentially reducing their stability within the active site. Additionally, the simulation does not account for dynamic conformational changes in the kinase, which may influence binding efficiency. Therefore, molecular dynamics simulations and in vitro kinase assays are necessary to validate these findings.

### 3.4. Molecular Dynamics and Free Energy

The molecular dynamics simulations demonstrated that Artepillin C maintained a stable and specific interaction with the enzyme B-Raf^600^E over a 500 ns timescale. The stabilization of the Artepillin-B-Raf complex, particularly from 200 ns onward, suggests favorable conformational accommodation and binding. The consistent RMSD and radius of gyration values indicate that the enzyme structure remains compact and stable in the presence of Artepillin C, while the free enzyme fluctuated more significantly within the same timeframe, reinforcing the hypothesis that ligand binding enhances structural stability while preserving structural integrity, a behavior also observed for Vemurafenib. The persistence of hydrogen bonds and the prominent ion-ion interaction with LYS473 suggest that electrostatic forces play a significant role in complex stability. Notably, this interaction appears to be a distinguishing feature between the docking and the most representative dynamics pose, with a repositioning of the interaction site from LYS483 (in docking) to LYS473 (in dynamics), possibly due to structural relaxation and optimization of the binding conformation. Artepillin C also reproduced some of the key interactions observed for Vemurafenib, including hydrogen bonds with ASP594 and a network of hydrophobic contacts within the ATP-binding pocket. While Vemurafenib relies on a π-cation interaction with LYS483, Artepillin’s stronger ionic interaction with LYS473 may provide an alternative mode of stabilization. Although LYS473 is not present in decomposition analysis as a contributing residue, it does not mean this interaction is not happening, as can be seen for the representative pose from the cluster analysis. Moreover, this strong interaction is not possible for Vemurafenib in its binding mode. Artepillin C also presents the slightest free energy of binding compared to that of Vemurafenib, mostly due to entropic loss differences. These free energy of binding interactions, combined with favorable docking energy and dynamic stability, support the hypothesis that artepillin C is a promising candidate for further investigation as a selective B-Raf^600^E inhibitor.

## 4. Materials and Methods

### 4.1. Physicochemical Properties and ADME Studies

The 2D structures of the phytochemicals selected for this study were designed using ChemDraw Ultra 8.0. The resulting compounds were then processed to obtain their SMILES representations, which were subsequently analyzed using the SwissADME Web tool (https://www.swissadme.ch/; accessed on 23 December 2024) [47] to evaluate their pharmacological and toxicological properties in silico. The data generated were used for further analysis.

### 4.2. Electronic Properties Calculations

The initial 2D structures were converted to their respective 3D projections, thus attributing the intra- and intermolecular interactions. Then, geometry optimization was performed using a molecular mechanics (UFF) force field using Avogadro 1.2.0v software [48]. Partial atomic charges were calculated employing Austin Model 1 (AM1) semi-empirical methodology, implemented in MOPAC2016 [49], and subsequently, the geometry optimization and electrostatic partial atomic charges (CHELPG) were computed using the ab initio method HF/6-31G* [49,50] implemented at ORCA 5.0.3 software [51]. To visualize the electronic properties of the molecules, including the frontier molecular orbitals (HOMO and LUMO) and the molecular electrostatic potential (MEP), a conversion process was necessary, as ORCA’s native output formats are not directly compatible with GaussView. After completion of the ORCA calculations, the binary wavefunction file (.gbw) was processed using the auxiliary tool (orca_2mkl), which generated a Molden input file (.molden.input). This file was then opened in the Molden software, where the molecular orbitals and electron density data could be further examined and exported in a format compatible with Gaussian [52].

### 4.3. Preparation of Target Protein

The crystal structure coordinates of human B-Raf Kinase V600E oncogenic mutant (BRAF V600E) (PDB ID: 3OG7) in the bound state with inhibitor Vemurafenib^®^ determined by X-ray diffraction were obtained from the Protein Data Bank [53] in PDB format. AutoDockTools 1.5.6 [54] were used for the preparation of the target. The protein structure was prepared for docking analysis by removing the compound and water molecules, attached compounds and ions. Polar hydrogen atoms and Kollman charges were added to the structure. Chain A of BRAF V600E was selected for docking study as the inhibitor Vemurafenib^®^ is in chain A [55].

### 4.4. Molecular Docking

The three-dimensional geometry models of B-Raf Kinase V600E Oncogenic Mutant were retrieved from the Protein Data Bank (PDB) (PDB ID: 3OG7, resolution 2.45 Å) and used as geometry reference to perform the redocking validation test. The target was prepared by the removal of heteroatoms (water, ions, and ligands) and by checking for potential structural errors, mainly those located at the recognition sites to be explored in this study. Molecular docking was performed employing AutoDock Tools-1.5.6 software. The compounds were then redocked considering a grid box with dimensions 40 × 40 × 40 and centered at the coordinates X = −1.385, Y = −12.941 and Z = −18.992. The redocking protocol was validated using the same software and target, employing Vemurafenib as the reference ligand. Fifty runs with ten resulting poses were generated and analyzed. The parameters of the Genetic Algorithm were kept as default in both studies, and the docking resulting poses were ordered according to their score values and analyzed using the freeware visualization program Discovery Studio Visualizer. All possible interactions and the steric complementarity obtained were analyzed considering a maximum distance of 3.0 Å to hydrogen bond interactions as well as minimum angles for hydrogen bond donors of 120 °C and for hydrogen bond acceptors of 90 °C. Interactions involving electronic π systems such as π-cation, π-stacking dipole-dipole and ion-dipole interactions were also considered, according to pre-settled criteria in the Discovery Studio Visualizer program.

### 4.5. Molecular Dynamics

Molecular dynamics studies were performed employing the best docking pose of Compound 3 into the target as the input file. The complex was prepared using CHARMM-GUI [56] web-based software considering a pH of 7.4 to calculate the amino acids ionization states, followed by the addition of the counterions Na^+^ and Cl^−^ to stabilize the charges. A cubic water box with a side length of 10 Å was constructed around the complex using the TIP3P water model, and periodic boundary conditions were added. The molecular dynamics simulation was accomplished using the GROMACS 2024 software [57]. The system underwent minimization using the steepest descent algorithm for 50.000 steps, reaching convergence at 1000 kJ/(mol·nm). For equilibration, NVT ensemble was applied at a temperature of 310.15 K for 5 ns using the v-rescale thermostat. Finally, the production phase was carried out using an NPT ensemble for 500 ns, under the temperature of 310.15 K, with the v-rescale thermostat, and at a pressure of 1 atm, maintaining the C-rescale barostat.

### 4.6. Free Energy of Binding

The binding affinities were calculated with the Molecular Mechanics Poission-Boltzmann Surface Area (MMPBSA) method for both Artepillin C and Vemurafenib complexes from 200 ns onwards, i.e., after observable stabilization. The software gmxMMPBSA [58] was employed. At first, the contributing residues were established within a radius of 6 Å from the ligands. The method, then, decomposes the free energy of binding as a function of the constituent residues of the protein. Therefore, E_MM_, G_polar_ and G_nonpolar_ energy components are computed for each individual atom of the contributing residues in each bound and unbound state. Thereafter, the procedure calculates the interaction energy contribution of the residue *x*, as demonstrated:$${∆R}_{x}^{BE}=\sum_{i=0}^{n}\left(A_{i}^{bound}-A_{i}^{free}\right)$$
where *A_i_^bound^* and *A_i_^free^* are the energy of atom *i* from residue *x*, and *n* stands for the total number of atoms in the residue. Therefore, the summation of the energy contribution from all residues is equivalent to the interaction energy, meaning ${∆G}_{binding}=\sum_{x=0}^{m}{∆R}_{x}^{BE}$, *m* being the total number of residues contributing to the complex. Yet only the enthalpic component Δ*H*, could be obtained using the gmxMMPBSA tool; for the entropic component –TΔS, quasiharmonic analysis was computed in gromacs. For that, a covariance matrix was computed to obtain the eigenvectors and eigenvalues. Vibrational entropy was then calculated:$$S=k_{B}\sum_{i}[ln\left(\frac{k_{B}T}{hv_{i}}\right)+1]$$
where k_B_ is the Boltzmann constant, T is temperature, *h* is the Planck constant, and *v_i_* is the frequency associated with the mode *i*.

## 5. Conclusions

This study highlights the promising potential of Artepillin C, a major bioactive compound from Brazilian green propolis, as a selective inhibitor of the oncogenic B-Raf^600^E kinase. Through a combination of molecular docking and long-scale molecular dynamics simulations, Artepillin C demonstrated favorable binding affinity, persistent interactions with key catalytic residues, and structural stability within the ATP-binding site. These computational results suggest that Artepillin C could interfere with the MAPK signaling pathway, offering a natural and potentially less toxic alternative for melanoma treatment. Furthermore, the in silico analysis of physicochemical and ADME properties reinforces its drug-likeness and pharmacokinetic viability. Taken together, these findings support the continued investigation of Artepillin C and related phytochemicals as scaffolds for the development of novel targeted therapeutics against melanoma driven by B-Raf^600^E mutation.

## Figures and Tables

**Figure 1 pharmaceuticals-18-00902-f001:**
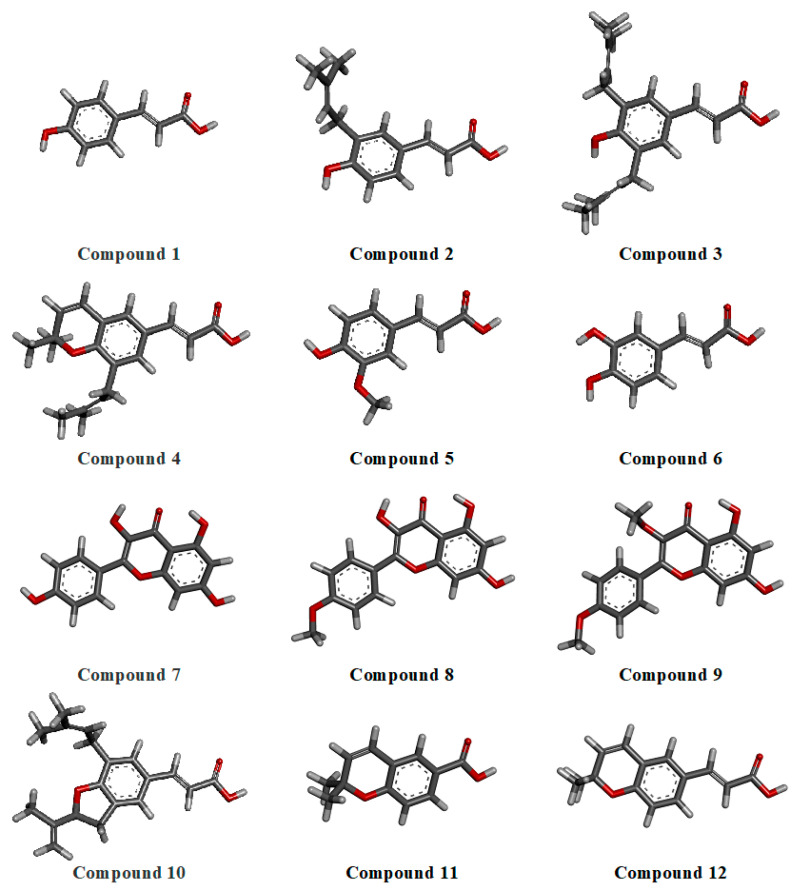
Optimized 3D structures of major phytochemicals found in Brazilian green propolis.

**Figure 2 pharmaceuticals-18-00902-f002:**
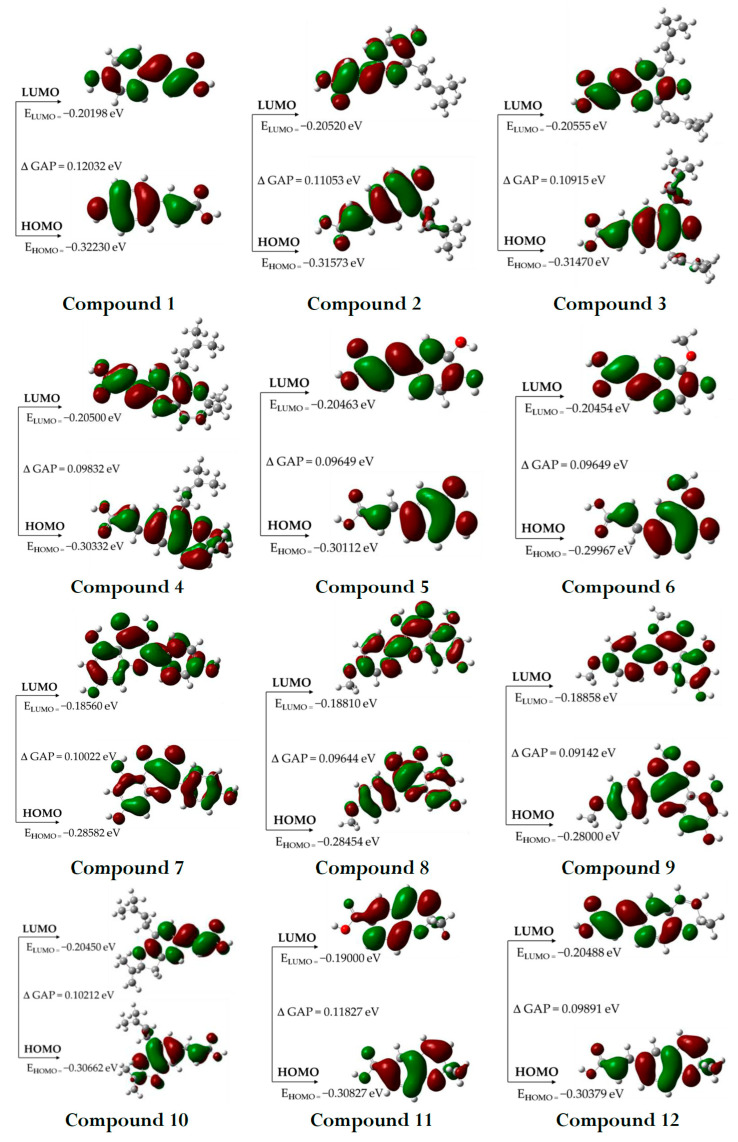
Frontier molecular orbitals (HOMO and LUMO) of the selected compounds.

**Figure 3 pharmaceuticals-18-00902-f003:**
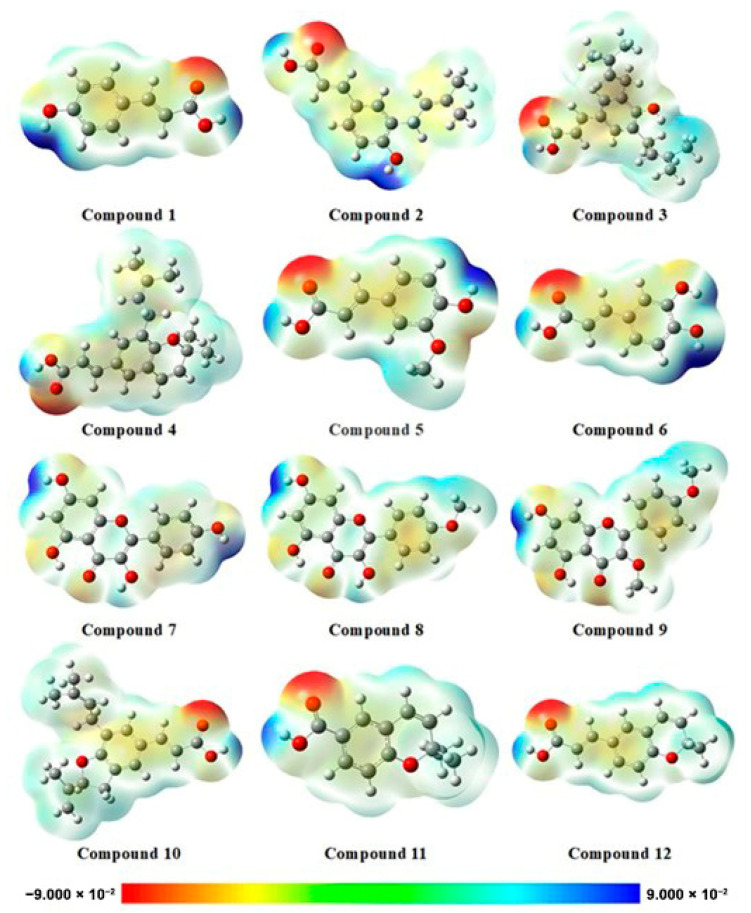
Electrostatic potential maps of the studied compounds highlighting reactive and interaction-prone regions.

**Figure 4 pharmaceuticals-18-00902-f004:**
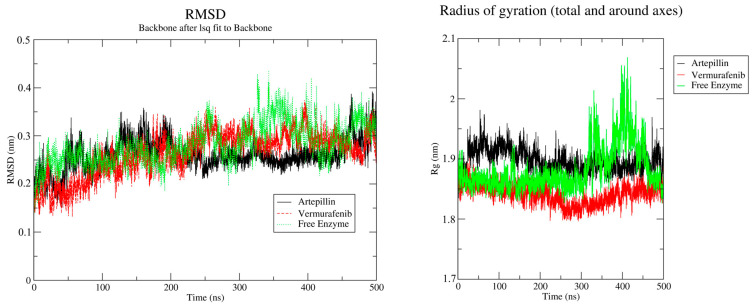
Radius of gyration and RMSD behavior along the molecular dynamic simulation time in ns. Red lines—Vemurafenib-enzyme complex simulation; Black lines—Artepillin C-enzyme complex simulation; Green lines—free enzyme simulation.

**Figure 5 pharmaceuticals-18-00902-f005:**
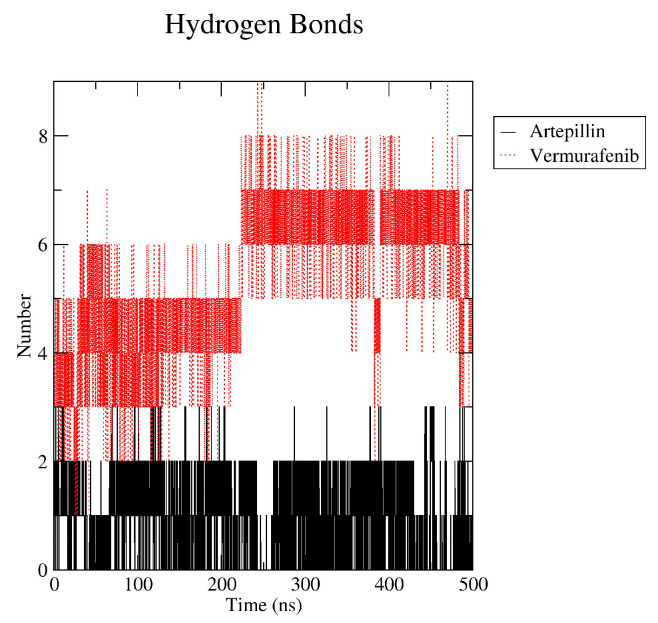
Hydrogen bond pattern along the molecular dynamic’s simulation.

**Figure 6 pharmaceuticals-18-00902-f006:**
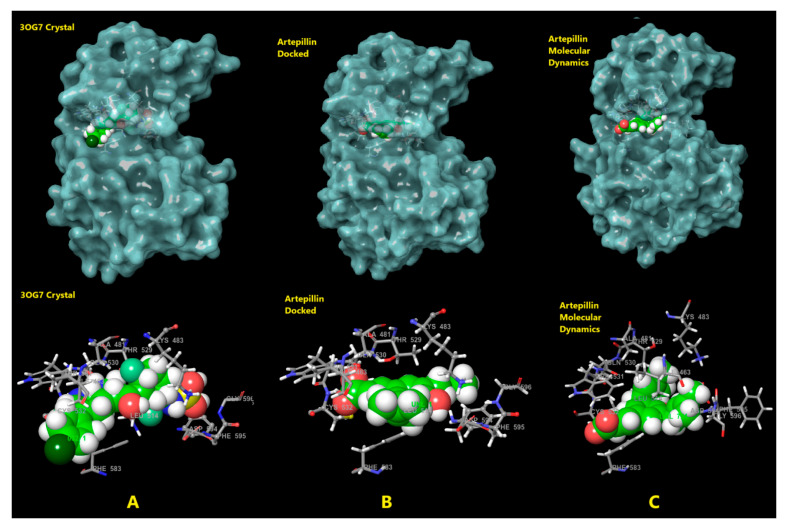
Molecular representation of the complexes and a zoomed-in view highlighting the closest residues of the protein. (**A**) Crystal structure from the input obtained in the presence of the ligand Vemurafenib; (**B**) Best-docked pose obtained to compound **3**; (**C**) Most representative pose obtained to Artepillin after molecular dynamics simulation.

**Figure 7 pharmaceuticals-18-00902-f007:**
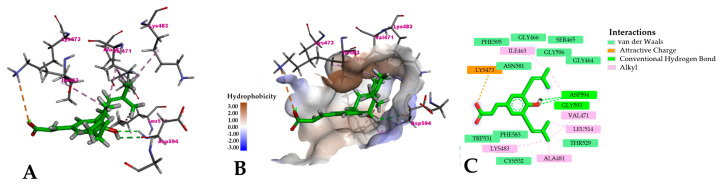
Interaction pattern of Artepillin C at the binding site. (**A**) Tridimensional view of the interactions; (**B**) Interactions located inside the hydrophobic surface map; (**C**) Two-dimensional view of the interactions.

**Figure 8 pharmaceuticals-18-00902-f008:**
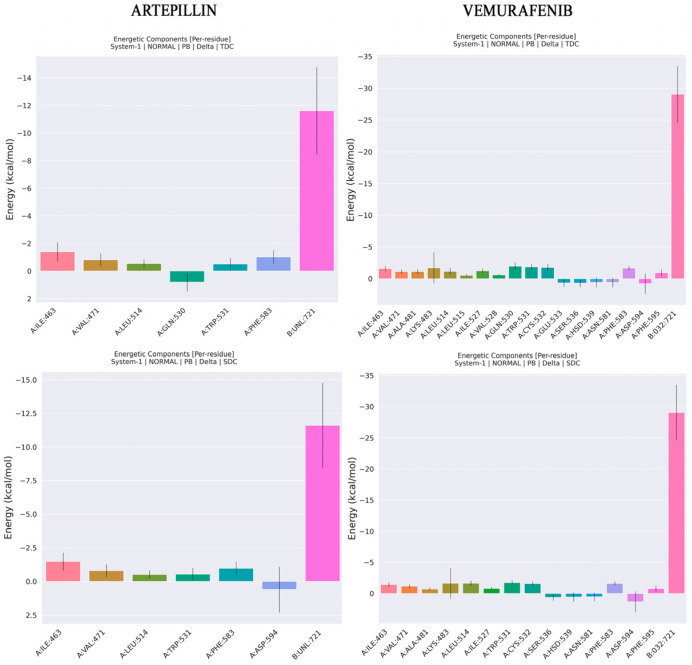
Decomposition interaction analysis for Artepillin C and Vemurafenib. TDC (Total decomposition), total interaction energy contribution for each residue. SDC (Sidechain decomposition) is the only sidechain contribution for interaction energy.

**Table 1 pharmaceuticals-18-00902-t001:** Calculated physicochemical properties of the phytochemicals considered in this study.

Ligand	MF	MW (g/mol)	nHA	nAHA	F.Csp^3^	nRB	nHBA	nHBD	MR	TPSA (Å^2^)
1	C_9_H_8_O_3_	164.16	12	6	0	2	3	2	45.13	57.53
2	C_14_H_16_O_3_	232.28	17	6	0.21	4	3	2	68.85	57.53
3	C_19_H_24_O_3_	300.39	22	6	0.32	6	3	2	92.57	57.53
4	C_19_H_22_O_3_	298.38	22	6	0.32	4	3	1	90.95	46.53
5	C_10_H_10_O_4_	194.18	14	6	0.1	3	4	2	51.63	66.76
6	C_9_H_8_O_4_	180.16	13	6	0	2	4	3	47.16	77.76
7	C_15_H_10_O_6_	286.24	21	16	0	1	6	4	76.01	111.13
8	C_16_H_12_O_6_	300.26	22	16	0.06	2	6	3	80.48	100.13
9	C_17_H_14_O_6_	314.29	23	16	0.12	3	6	2	84.95	89.13
10	C_19_H_22_O_3_	298.38	22	6	0.32	6	3	1	90.12	46.53
11	C_12_H_12_O_3_	204.22	15	6	0.25	1	3	1	57.52	46.53
12	C_14_H_14_O_3_	230.26	17	6	0.21	2	3	1	67.23	46.53

MF—Molecular Formula; MW (g/mol)—Molecular weight; nHA—Number of heavy atoms; nAHA—Number of aromatic heavy atoms; F.Csp^3^—Fraction of sp^3^ hybridized carbon atoms; nRB—Number of rotatable bonds; nHBA—Number of hydrogen bond acceptors; nHBD—Number of hydrogen bond donors; MR—Molar refractivity; TPSA (Å^2^)—Topological polar surface area. Ligands—*p*-coumaric acid (**1**), drupanin (**2**), Artepillin C (**3**), culifolin (**4**), ferulic acid (**5**), caffeic acid (**6**), kaempferol (**7**), kaempferide (**8**), ermanin (**9**), (*E*)-2-isopropenyl-7-isopentenyl-2,3-dihydrobenzofuran-5-acrylic acid (**10**), 2,2-dimethyl-2*H*-1-benzopyran-6-carboxylic acid (**11**), and(*E*)-3-(2,2-dimethyl-2*H*-1-benzopyran-6-yl) propenoic acid (**12**).

**Table 2 pharmaceuticals-18-00902-t002:** Predicted ADME properties of ligands based on in silico pharmacokinetic analysis.

Ligand	LogP	GI	BBB	P-gp Substrate	Cytochrome Inhibitor P450				
					CYP1A2	CYP2C19	CYP2C9	CYP2D6	CYP3A4
1	1.26	High	Yes	No	No	No	No	No	No
2	2.83	High	Yes	No	No	No	No	No	No
3	4.27	High	Yes	No	No	Yes	Yes	No	No
4	4.02	High	Yes	No	No	Yes	Yes	Yes	No
5	1.36	High	Yes	No	No	No	No	No	No
6	0.93	High	No	No	No	No	No	No	No
7	1.58	High	No	Yes	No	No	Yes	Yes	No
8	2.00	High	No	Yes	No	No	Yes	Yes	No
9	2.35	High	No	Yes	No	Yes	Yes	Yes	No
10	4.11	High	Yes	Yes	Yes	Yes	No	Yes	Yes
11	2.33	High	Yes	Yes	No	No	No	Yes	No
12	2.67	High	Yes	Yes	Yes	No	No	Yes	Yes

LogP—Measures lipophilicity, affecting membrane permeability; GI—Indicates absorption in the gastrointestinal tract; BBB—Shows ability to cross the blood–brain barrier; P-gp Substrate—Identified by P-glycoprotein, influencing drug transport and distribution. Ligands: *p*-coumaric acid (**1**), drupanin (**2**), Artepillin C (**3**), culifolin (**4**), ferulic acid (**5**), caffeic acid (**6**), kaempferol (**7**), kaempferide (**8**), ermanin (**9**), (*E*)-2-isopropenyl-7-isopentenyl-2,3-dihydrobenzofuran-5-acrylic acid (**10**), 2,2-dimethyl-2*H*-1-benzopyran-6-carboxylic acid (**11**), and (*E*)-3-(2,2-dimethyl-2*H*-1-benzopyran-6-yl) propenoic acid (**12**).

**Table 3 pharmaceuticals-18-00902-t003:** Physicochemical properties of the phytochemicals considered in this study.

Ligand	Binding Energy (kcal/mol)	Hydrogen Interacting Residue of Target Along with Their Bond Length	Hydrophobic Interacting Residue of Target Along with Their Bond Length
1	−5.11	CYS532 (2.01), GLY534 (1.88), GLN530 (1.72)	TRP531, CYS532
2	−7.02	CYS532 (1.76), ILE527 (2.62), GLN530 (2.05)	THR529, PHE595, LEU505, LEU505, LEU514, ALA481, LYS483
3	−8.17	LYS483 (2.09), CYS532 (2.07), GLN530 (2.02)	VAL471, ILE463, VAL471, LEU505, LEU505, LEU514, PHE468, PHE595, VAL471
4	−7.57	CYS532 (1.85)	CYS532, PHE583, VAL471, ALA481, ALA481, LEU514, VAL471, LYS483, VAL471, LYS483, VAL471, PHE468, PHE583, PHE583, VAL471, ALA481
5	−5.51	PHE595 (2.94)	GLY596, ALA481, VAL471, VAL471, ALA481, LYS483
6	−5.41	LYS483 (2.62), PHE595 (2.59), GLY596 (2.13), ALA481 (2.21), ILE527 (2.04)	VAL471, ALA481, LYS483
7	−7.15	THR529 (2.59), CYS532 (1.77), ILE527 (2.04), GLN530 (2.03), CYS532 (2.29)	TRP531, TRP531, PHE583, PHE583, VAL471, ALA481, CYS532, ILE463, VAL471, VAL471, ALA481, LYS483
8	−7.38	LYS483 (2.34), CYS532 (1.69), GLN530 (1.98), CYS532 (2.23)	TRP531, TRP531, PHE583, PHE583, VAL471, ALA481, CYS532, ILE463, LYS483, LEU514
9	−7.29	CYS532 (1.71), GLN530 (2.04)	ASP594, LYS483, TRP531, LEU505, VAL471, ALA481, VAL471, ALA481, CYS532, LYS483
10	−7.74	LYS483 (1.73), PHE595 (2.99), GLY596 (1.96)	VAL471, ALA481, ALA481, LEU514, CYS532, VAL471, TRP531, TRP531, TRP531, PHE583, VAL471, ALA481, LYS483, LEU514
11	−6.74	LYS483 (1.80), ASP594 (2.22), PHE595 (2.99), GLY596 (2.24)	LYS483, ALA481, ALA481, ALA481, LEU514, LEU514, VAL471, PHE583, LYS483, LEU514
12	−8.49	LYS483 (1.70), ASP594 (2.29), PHE595 (2.99), GLY596 (2.21), PHE583 (2.85)	VAL471, ALA481, ALA481, LEU514, CYS532, VAL471, VAL471, ILE463, VAL471, PHE468, TRP531, TRP531, PHE583, PHE583, VAL471, LEU514

Ligands: *p*-coumaric acid (**1**), drupanin (**2**), Artepillin C (**3**), culifolin (**4**), ferulic acid (**5**), caffeic acid (**6**), kaempferol (**7**), kaempferide (**8**), ermanin (**9**), (E)-2-isopropenyl-7-isopentenyl-2,3-dihydrobenzofuran-5-acrylic acid (**10**), 2,2-dimethyl-2H-1-benzopyran-6-carboxylic acid (**11**), and (E)-3-(2,2-dimethyl-2H-1-benzopyran-6-yl) propenoic acid (**12**).

**Table 4 pharmaceuticals-18-00902-t004:** Free energy of biding from both Artepillin C and Vemurafenib.

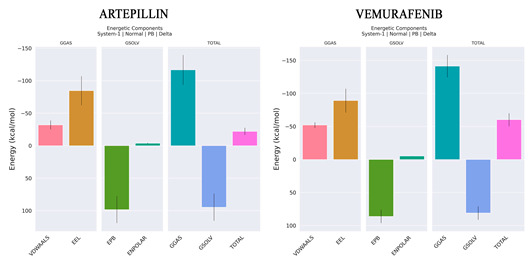		
ΔE (VDW)	−32.03	−52.13
ΔE (EEL)	−84.62	−89.14
ΔE (EPB)	98.26	86.12
ΔE (ENPOLAR)	−3.65	−5.1
GGAS	−116.65	−141.27
GSOLV	94.6	81.02
ΔH (total)	−22.05	−60.25
−TΔS	1.28	42.70
ΔG	−20.77	−17.55

The components are defined as ΔE (VDW), the energy from van der Waals interactions, and ΔE (EEL), the energy from electrostatic interactions. ΔE (EPB) is the energy from polar solvation, and ΔE (ENPOLAR) is the repulsive component of the nonpolar solvation energy. The total free energy in the gas phase, GGAS, is calculated as GGAS = ∆EVDW + ∆EEL. The total solvation-free energy, GSOLV, is determined as GSOLV = ∆EPB + ∆ENPOLAR. Other terms used are ΔH for enthalpy of binding; −TΔS for conformational entropy loss contribution upon ligand binding; and ΔG (binding), represented as ΔH–TΔS.

## Data Availability

The original contributions presented in the study are included in the article, further inquiries can be directed to the corresponding author.

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
