# Peer review of "Exploring Brazilian Green Propolis Phytochemicals in the Search for Potential Inhibitors of B-Raf600E Enzyme: A Theoretical Approach"

_pharmaceuticals, 2025, doi:10.3390/ph18060902_

Round 1
Reviewer 1 Report
Comments and Suggestions for Authors
The provided manuscript entitled: "Exploring Brazilian Green Propolis Phytochemicals in the Search for Potential Inhibitors of B-Raf⁶⁰⁰E Enzyme: A Theoretical Approach" by Goncalves presents a comprehensive computational study of 12 isolated compounds including p-coumaric acid (1), drupanin (2), Artepillin C (3), culifolin (4), ferulic acid (5), caffeic acid (6),
kaempferol (7), kaempferide (8), ermanin (9), (E)-2-isopropenyl-7-
isopentenyl-2,3-dihydrobenzofuran-5-acrylic acid (10), 2,2-dimethyl-2H-1-benzopyran-6-
carboxylic acid (11), and (E)-3-(2,2-dimethyl-2H-1-benzopyran-6-yl)propenoic acid (12) from Brazilian green propolis as potential inhibitors of the B Raf⁶⁰⁰E mutant protein.
The manuscript needs major revisions.
- Typo errors: Ligand and Compound, the first characters “L’ and “C’ in normal, not capital. Check entire manuscript.
- Page 9/20: Correct Green Propolis as green propolis.
Table 1 as Table 3
- Provide the binding results of vemurafenib and validation of docking results.
- Part 4: The discussion is lengthy. Write the part 4.1, 4.2.1 and 4.2.2 concisely.
- Remove the sentence : Essa característica pode influenciar sua estabilidade metabólica e biodisponibilidade, reduzindo a suscetibilidade ao metabolismo oxidativo e possivelmente prolongando sua meia-vida biológica.
- Correct the references in page 10: 45, 56, 57
- Provide the results of molecular dynamic simulations of the remained 11 compounds and vemurafenib.
Validation the theoretical results by in-vitro kinase assay.
Author Response
We sincerely thank you for the thorough and constructive review of our manuscript entitled ‘Exploring Brazilian Green Propolis Phytochemicals in the Search for Potential Inhibitors of B-Raf^V600E Enzyme: A Theoretical Approach’. The suggestions and comments have certainly contributed significantly to improving the quality of our work. Below, we provide detailed responses to each observation
Reviewer Comments and Responses:
REVIEWER 1:
1. Typo errors: Ligand and Compound, the first character’s “L” and “C” in normal, not capital. Check the entire manuscript.
Response: We corrected the use of "Ligand" and "Compound" throughout the entire manuscript. These terms now appear in lowercase.
2. Page 9/20: Correct "Green Propolis" to "green propolis". Also, Table 1 should be Table 3.
Response: "Green Propolis" has been corrected to "green propolis" throughout the manuscript. The table label inconsistency has been corrected; the docking results are now correctly referenced as Table 3.
3. Provide the binding results of vemurafenib and validation of docking results.
Response: We sincerely apologize for the confusion in the previous version of the manuscript. There was a misunderstanding in the way the docking validation procedure was described. To clarify, the redocking protocol was indeed performed using Vemurafenib, the co-crystallized ligand of the B-Raf^V600E enzyme, as the reference compound. This procedure aimed to assess the reliability of the docking parameters by verifying the ability of the software to reproduce the experimentally observed binding pose. We have now included the redocking results of Vemurafenib in the Materials and Methods’ section.
4. The discussion is lengthy. Write sections 4.1, 4.2.1, and 4.2.2 more concisely.
Response: We revised these sections for conciseness while preserving scientific rigor. The descriptions of physicochemical parameters, electronic properties, and molecular orbital analyses were streamlined, focusing only on the most relevant data supporting the docking and dynamics results.
5. Remove the sentence in Portuguese: "Essa característica pode influenciar sua estabilidade metabólica e biodisponibilidade, reduzindo a suscetibilidade ao metabolismo oxidativo e possivelmente prolongando sua meia-vida biológica."
Response: The sentence in Portuguese was indeed an oversight and has been removed entirely.
6. Correct the references in page 10: 45, 56, 57.
Response: References 45, 56, and 57 were revised to correct formatting errors and mismatches. All references were checked against the bibliography, and corrections were applied to both the numbering and the reference list to ensure consistency.
7. Provide the results of molecular dynamics simulations for the remaining 11 compounds and vemurafenib.
Response: We appreciate the reviewer’s comment and would like to clarify that the MD was not performed for all 11 compounds. As originally planned and now described, MD simulations were conducted only for the Vemurafenib–B-Raf^V600E complex to validate the docking protocol and assess the stability of the reference ligand. This request has been fully addressed, and the MD results for Vemurafenib are now properly included in the revised manuscript. In response to this, the MD section has been rewritten, including the updated methodology and the results obtained from the simulation.
We would like to clarify that this study was entirely designed as a theoretical and computational approach, with the primary objective of evaluating the potential of Brazilian green propolis phytochemicals as B-Raf^V600E inhibitors through in silico methods. No experimental (in vitro or in vivo) assays were planned or conducted as part of this work. The purpose of this study is to provide preliminary insights that can guide and prioritize future experimental investigations. Therefore, it is not possible to include in vitro validation results in the present manuscript.
We would like again to thank the reviewers for their valuable contributions, constructive criticism and suggestions, which have undoubtedly resulted in significant improvements in the scientific quality and organization of our work. We look forward to the final evaluation and the possible approval of our manuscript for publication.
Sincerely,
The authors.
Reviewer 2 Report
Comments and Suggestions for Authors
Dear authors!
I have studied your research with great interest, the potential of this research is very large and, if fully implemented, has the broadest prospects.
This study highlights the promising potential of Artepillin C, a major bioactive compound from Brazilian green propolis, as a selective inhibitor of the oncogenicB-Raf-E kinase. Through a combination of molecular docking and long-scale molecular dynamics simulations, Artepillin C demonstrated favorable binding affinity, persistent interactions with key catalytic residues, and structural stability within the ATP-binding site. These computational results suggest that Artepillin C could interfere with the MARK signaling pathway, offering a natural and potentially less toxic alternative for melanoma treatmentHowever, as we studied the manuscript, some questions arose that require clarification.
- In Table 1, the authors provide chemical formulas of the compounds with the greatest potential, isolated from Brazilian propolis, which are then manipulated. One of the compounds, judging by the text, is Artepillin C, the other 11 compounds listed in the table are not named. It is necessary to provide the names of the compounds being studied and at least brief explanations of what the choice was based on: previous studies, studies by other authors, etc. The fact is that propolis is very rich in bioactive compounds, about several hundred, and readers need to understand what chemical compounds are being discussed.
- Authors write: The structural analysis indicates that molecules with high aromaticity and TPSA, such as Ligands 7,8,9 are likely to exhibit strong hydrogen bonding interactions, influencing their bioactivity and selectivity. And then follows the phrase: Molecules with moderate Fraction CSp3, such as Ligand 3, demonstrated a balanced profile of rigidity and flexibility, which could optimize receptor binding and pharmacokinetic properties. So what do the authors consider most important for constructing a future medicinal formula? The properties for Ligands 7,8,9? or those inherent to Ligand 3?
Author Response
We sincerely thank you for the thorough and constructive review of our manuscript entitled ‘Exploring Brazilian Green Propolis Phytochemicals in the Search for Potential Inhibitors of B-Raf^V600E Enzyme: A Theoretical Approach’. The suggestions and comments have certainly contributed significantly to improving the quality of our work. Below, we provide detailed responses to each observation.
Reviewer Comments and Responses:
REVIEWER 2:
1. In Table 1, the authors provide chemical formulas but no names. Include the names of the compounds and explain the basis for their selection.
Response: We appreciate the reviewer’s suggestion. However, due to the space constraints and formatting requirements of the journal, it was not possible to include the compound names directly within the table in an adequate and clear manner. Nevertheless, we fully understand the importance of this information and have addressed the suggestion by including the names of the compounds in the figure captions of each table where they are discussed. A paragraph was added in Section 1 (Introduction) explaining that these compounds were selected based on previous phytochemical studies of Brazilian green propolis.
2. Clarify what structural features are most important for future medicinal formulation: high aromaticity and TPSA (e.g., ligands 7,8,9) or moderate Fraction Csp³ (ligand 3)?
Response: Following the restructuring of the text suggested by the other reviewer, this text had to be completely rewritten to ensure greater clarity and the text was incorporated in a way that was adjusted to the new context of the manuscript.
We would like again to thank the reviewer for their valuable contributions, constructive criticism and suggestions, which have undoubtedly resulted in significant improvements in the scientific quality and organization of our work. We look forward to the final evaluation and the possible approval of our manuscript for publication.
Sincerely,
The authors.
Round 2
Reviewer 1 Report
Comments and Suggestions for Authors
The authors have modified and corrected as the reviewer requested. The paper can be accepted with minor revisions.
- Compound numbers in bold. Check the entire manuscript.
- Correct Drupanin (2) [15], Culifolin (4), Ermanin (9) as drupanin (2) [15], culifolin (4), ermanin (9)
Author Response
Dear Reviewer,
We would like to sincerely thank you for your suggestions for the improvement of our manuscript. All the modifications requested have been carefully addressed. Specifically, we corrected the formatting of compound names, changing Drupanin (2) [15], Culifolin (4), Ermanin (9) to drupanin (2) [15], culifolin (4), ermanin (9) throughout the entire manuscript. Additionally, compound numbers have been formatted in bold, as suggested. All changes made in response to your comments have been highlighted in yellow in the revised manuscript for your convenience.
We greatly appreciate your time and consideration.
Sincerely,
The authors.